

# Location, biophysical and agronomic parameters for croplands in Northern Ghana

Jose Luis Gómez-Dans[1,2], Philip Lewis[1,2], Feng Yin[1,2], Kofi Asare[3], Patrick Lamptey[3], Kenneth Kobina Yedu Aidoo[3], Dilys MacCarthy[4], Hongyuan Ma[2], Qingling Wu[2], Martin Addi[3], Stephen Aboagye-Ntow[3], Caroline Edinam Doe[3], Rahaman Alhassan[5], Isaac Kankam-Boadu[5], Jianxi Huang[6], and Xuecao Li[7]

[1]National Centre for Earth Observation, UK
[2]Dept. of Geography, University College London, UK
[3]Ghana Space Science Technology Institute, Accra, Ghana
[4]Soil and Irrigation Research Centre, University of Ghana, Accra, Ghana
[5]ADRA Ghana, Tamale, Ghana
[6]China Agricultural University, Beijing, China
[7]Key Laboratory of Remote Sensing for Agri-Hazards, Ministry of Agriculture and Rural Affairs, Beijing 100083, China

**Correspondence:** Jose L Gómez-Dans (jgomezdans@gmail.com)

**Abstract.** Smallholder agriculture is the bedrock of the food production system in sub-Saharan Africa. Yields in Africa are significantly below potentially attainable yields for a number of reasons, and they are particularly vulnerable to climate change impacts. Monitoring of these highly heterogeneous landscapes is needed to respond to farmer needs, develop appropriate policy and ensure food security, and Earth Observation (EO) must be part of these efforts. There is a lack of ground data for developing and testing EO methods in West Africa, and in this paper, we present data on (i) crop locations, (ii) biophysical parameters and (iii) crop yield and biomass was collected in 2020 and 2021 in Ghana and is reported in this paper. In 2020, crop type was surveyed in more than 1800 fields in three different agro-ecological zones across Ghana (Guinea Savannah, Transition and Deciduous zones). In 2021, a smaller number of fields were surveyed in the Guinea Savannah zone, and additionally, repeated measurements of leaf area index (LAI) and leaf chlorophyll concentration were made on a set of 56 maize fields. Yield and biomass were also sampled at harvesting. LAI in the sampled fields ranged from 0.1 to $5.24\,\mathrm{m^2m^{-2}}$, whereas leaf chlorophyll concentration varied between 6.1 and $60.3\,\mathrm{\mu gcm^{-2}}$. Yield varied between 190 and $4580\,\mathrm{kg\,ha^{-1}}$, with an important within-field variability (average per field standard deviation $381\,\mathrm{kg\,ha^{-1}}$). The data are used in this paper to: (i) evaluate the Digital Earth Africa 2019 cropland masks where $61\,\%$ of sampled 2020/21 cropland is flagged as cropland by the data set; (ii) develop and test an LAI retrieval method from Earth Observation Planet surface reflectance data (validation correlation coefficient $R = 0.49$, RMSE $0.44\,\mathrm{m^2m^{-2}}$; (iii) create a maize classification dataset for Ghana for 2021 (overall accuracy within the region tested: 0.84); and (iv) explore the relationship between maximum LAI and crop yield using a linear model (correlation coefficient $R = 0.66$ and $R = 0.53$ for *in situ* and Planet-derived LAI, respectively). The data set, made available here within the context of the GEOGLAM initiative, is an important contribution to understanding crop evolution and distribution in smallholder farming systems, and will be useful for researchers developing/validating methods to monitor these systems using



Earth Observation data. The data described in this paper are available from https://doi.org/10.5281/zenodo.6632083 (Gomez-Dans et al., 2022).

# 1   Introduction

Agricultural production in Sub-Saharan Africa is dominated by smallholder farms that support most households (Giller et al., 2021; Antonaci et al., 2014). In Ghana, agriculture contributes around $20\,\%$ of GDP and employs around half of the population
(MOFA, 2010). Maize accounts for more than half of the country's cereal production (Ragasa et al., 2014), and in the North of the country, the crop is grown in rain-fed conditions, with low inputs (limited use of fertiliser, low uptake of modern/hybrid varieties, low mechanisation), suffering additionally from considerable post-harvesting losses and nutrient-poor soils (Freduah et al., 2019; MacCarthy et al., 2017; Sánchez, 2010),(MOFA, 2010). These factors result in an important yield gap compared to potential attainable yields (van Loon et al., 2019; Cairns et al., 2013). This yield gap is exacerbated in a climate change context,
where agricultural production in Ghana is likely to be further limited by increased temperature and more erratic rain regimes (Sultan and Gaetani, 2016; Chemura et al., 2020), with complex relationships with nitrogen use (Falconnier et al., 2020) and other social factors (Nyantakyi-Frimpong and Bezner-Kerr, 2015), adding further vulnerability to yields in the region.

Timely monitoring of smallholder maize production is important to understand the developing food security situation, but also to provide information to food producers and other value creators. Decision-makers at regional or national levels need this
for planning policy, import-export requirements, or other advance planning or support mechanisms for farmers (United Nations, 2013; Nakalembe et al., 2021). Monitoring capabilities are also important for developing crop insurance and maximising the economic potential (and hence livelihoods) of smallholder farmers (Benami et al., 2021). At more local scales, such information can be used by extension workers to assist farmers in improving their practice. (Carletto et al., 2013) argues that a lack of good quality agricultural data in Africa has hampered innovation and growth in this crucial economic sector.

In Ghana, $60\,\%$ of farms have an area of less than $1.2\,\mathrm{ha}$ (MOFA, 2010), giving rise to a highly heterogeneous landscape. Some factors that explain this heterogeneity are also common with the yield gap: limited access to e.g., irrigation, mechanisation and fertilser use, workforce scarcity, low labour productivity, limited access to finance, etc. (van Loon et al., 2019). Local patterns of crop yield are known to be impacted by local soil and meteorological conditions as well as farmer choices. For example, Freduah et al. (2019) note that maize planting occurs over a three month period, with further crop development variation
depending on the use of fertiliser and other management practices. This, alongside the varying quality of seed inputs, results in very different inter- and intra-field crop evolution even for crops subject to very similar weather patterns. This complicates both monitoring and modelling efforts.

There is a strong need for better information on cropland area, crop productivity, and the factors that affect crop production in Africa. Earth Observation (EO) has been shown to provide a practical data source to monitor croplands over large areas
for much of the world and contribute to the collection of better agronomic statistics. This in turn can be used by decision-makers, agronomists and other parties to better understand and plan actions to improve food security and alleviating poverty (e.g. (Baruth et al., 2008; Carletto et al., 2015; Brown, 2016)).



To monitor agriculture, the first layer of information is mapping the areas of croplands, to distinguish them from other land uses, but even this basic information is currently very uncertain for large parts of Ghana. To be able to estimate total production or even average productivity in some region, an additional level of sophistication is needed on top of that where the crop type is identified. While there have been some recent advances on cropland masks derived from Earth Observation for the region (Burton et al., 2022; Estes et al., 2021; Xiong et al., 2017), accurate, timely data sets that allow the location of individual crops over large areas are still mainly lacking.

The advent of frequent, medium- to high-resolution Earth Observation (EO) data over the last $\sim 5$ years from sensors such as Sentinel 2 (Drusch et al., 2012) and the Planet constellation (Planet, 2018) allows repeated data acquisitions over the growing season that give the potential to infer the status of crops at the field and sub-field level, even for smallholders. This is currently mainly done using empirical relationships between satellite-derived indicators (such as spectral vegetation indices) and *in situ* measurements of yield and/or above ground biomass. Typical approaches relate either the maximum value or the time integral of the signal over the season to yield or biomass (Becker-Reshef et al., 2010; Kouadio et al., 2012; Unganai and Kogan, 1998; Mkhabela et al., 2011; Franch et al., 2015; Petersen, 2018). The EO data act as indicators of green leaf biomass or green-up or senescence rates.

There is a concerted effort to provide a minimal set of so-called "Essential Agricultural Variables" (EAVs) that are required to monitor agriculture globally (Whitcraft et al., 2019). Some of the EAVs are more directly related to the status of the crop, e.g. leaf area index (LAI), the fraction of photosynthetically active radiation absorbed by the canopy (fAPAR), soil moisture, above ground biomass and leaf pigment concentrations such as chlorophyll concentration. Some of these biophysical parameters have been successfully derived from EO data (Verrelst et al., 2019). The derivation of biophysical parameters is more involved than calculating a vegetation index, but simplify the interpretation of EO data, reducing the effect of extrinsic/nuisance processes in the EO signal, such as soil colour or brightness variations, acquisition geometry effects, different sensor spectral configurations, etc. As the derivation is indirect, careful validation and assessment of uncertainties in the inference of these parameters is critical (Loew et al., 2017).

LAI is an important indicator of crop development, and its use for yield estimation has been shown superior to using simple indices (Baez-Gonzalez et al., 2005; Lambert et al., 2018). Although the relationship between surface reflectance and LAI is complex, it is possible to develop empirical mappings between LAI and vegetation indices, although these mappings may not be very general. Having estimates of LAI allows us to leverage mechanistic crop growth models to e.g., train the relationship between modelled LAI (as a function of meteorological data, typical management and soils) EO observations (Jin et al., 2017, 2019; Azzari et al., 2017; Jain et al., 2016), or more sophisticated data assimilation methods, based on combining the uncertain evidence from the model predictions with the (incomplete) EO-derived observations of LAI (Huang et al., 2019).

Ultimately, all of these approaches rely on *in situ* data for method development and validation. The need for new data collection is particularly acute for smallholder croplands in Sub-Saharan Africa, as most studies have concentrated in croplands and crops in the global North (Pritchard et al., 2022). The contribution of this paper is to provide and describe a data set covering three main aspects: (i) location of crops; (ii) biophysical parameters over the growing season for maize; and (iii) crop yield and biomass data. We demonstrate the application of the crop type/location data to validate cropland data sets and to train a maize





classifier using Sentinel 2 observations. This data set can be used to understand crop dynamics over three different regions. We describe a biophysical parameter and yield data set collected over a set of maize fields in northern Ghana between July and November 2021. An associated crop type data set for Ghana is also introduced, covering the same area as the biophysical parameters for the 2021 season, but with a wider-area mapping undertaken during 2020.

Estimating biophysical parameters from EO data is an indirect problem, and *in situ* data is needed to validate EO retrievals for particular environments. For croplands, tracking the vegetation over the entire growing season is particularly important. Here, we concentrate on LAI and leaf chlorophyll concentration as the parameters of interest. LAI was selected for its known relationship to yield and the use of LAI as a critical state variable in crop growth models. Leaf chlorophyll content has also been related to GPP (Gitelson et al., 2006) and yield (Croft et al., 2020). Leaf chlorophyll has also been linked to the CO2-saturated photosynthetic rate (Vmax) (Wang et al., 2021), which would provide an additional linkage into crop models to LAI, as well as potential information on nitrogen stress. The biophysical parameter data set is enhanced by collecting additional data on grain yield and biomass, both fundamental to understand food production and to allow the development of models that link EO data to this.

Releasing the data set described in this paper is a contribution to data sharing efforts championed by the Group on Earth Observations Global Agricultural Monitoring (GEOGLAM, https://www.earthobservations.org/geoglam.php), an initiative launched by the G20 international forum in 2011 (Becker-Reshef et al., 2020). The crops in the fields surveyed in this data set are grown by smallholder farmers and represent a typical sample of the variability found in this region. The data is available from https://doi.org/10.5281/zenodo.6632083 (Gomez-Dans et al., 2022).

## 2  Materials and methods

### 2.1  Location

In this study we present a new dataset of farm boundaries and biophysical parameter measurements in Ghana. The dataset includes information on crop type, collected in an extensive campaign covering areas of 50 km by 50 km in three agroecological zones across the country in December 2020 (see Fig. 1 for locations), and an intensive maize-focused campaign in 2021 in the North of Ghana, specifically the Northern and Savannah regions (see Fig. 2).

In the intensive campaign, data were collected from the Tamale, Mion, Salaga North, Gushegu, Karaga and Nanton districts. The intensive study area covers around 55 km East to West and close to 70 km North to South. It is within the Guinea Savannah agro-ecological zone of the country. Rainfall patterns in the area are unimodal, with a rainy season starting in April/May and ending in September/October. Mean daily temperatures oscillate between $31°$ in the hottest month (February/March) and around $22°$ in the coolest month (August/September). Annual rainfall ranges between 900 mm and 1100 mm (see Fig. 3). During the rainy season, when most crops are grown, cloud cover is persistent, making it often difficult to acquire observations using optical EO sensors. Additional difficulties for remote sensing of crops in this area include the prevalence of trees within fields, inter-cropping practices, and often the presence of significant weed cover early in the season, all of which complicate the interpretation of EO signals (see Fig. 4 for an example of this).

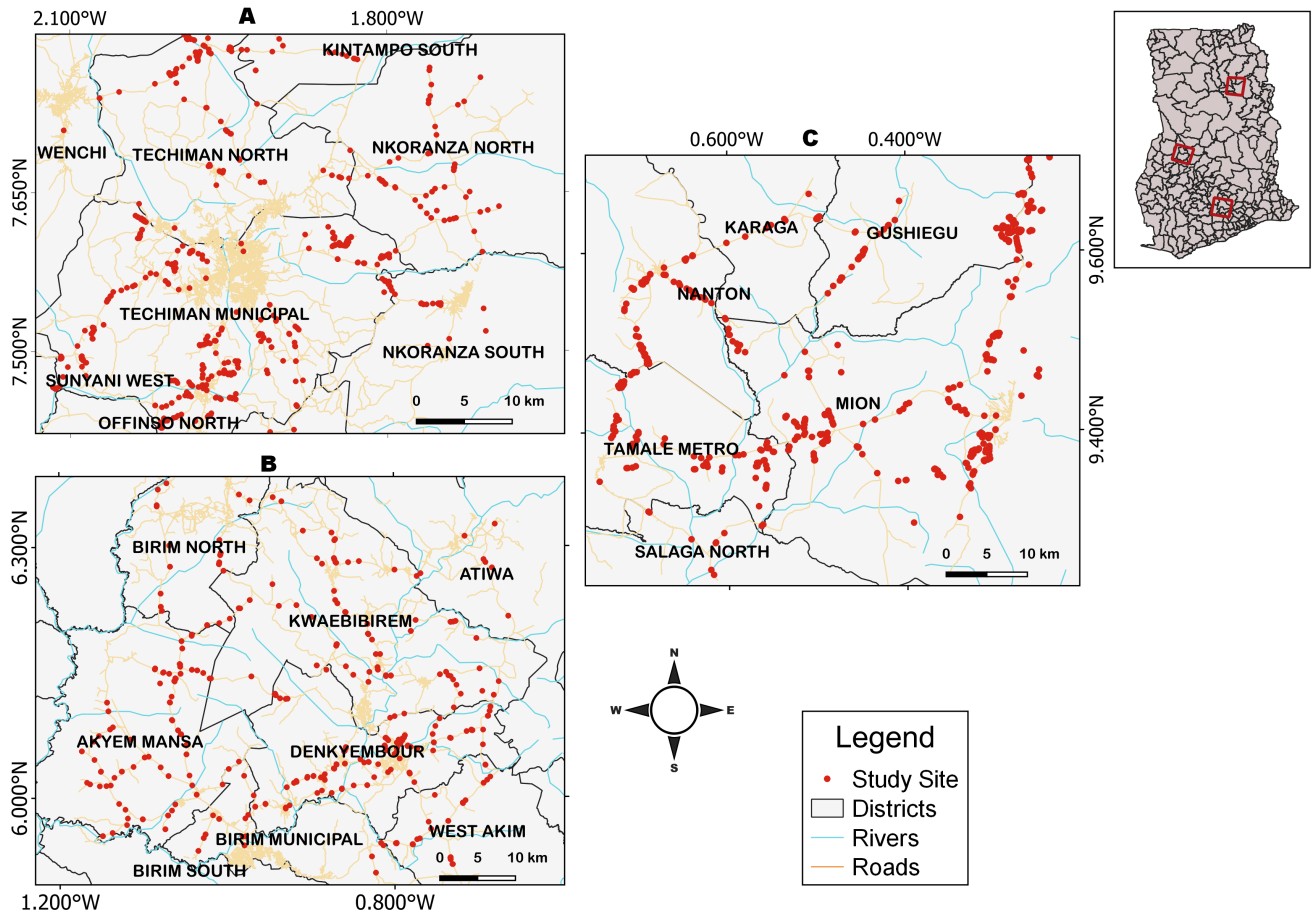

**Figure 1.** Location of samples for crop type mapping in the 2020 extensive campaign. (A) Transition zone (B) Deciduous and (C) Northern Savanah sites. Top right sub-image shows the location of these areas within Ghana.

## 2.2 Crop type mapping

Two crop type mapping campaigns were conducted: a preliminary campaign in December 2020, covering three agroecological zones, and a maize-focused campaign in 2021.

The 2020 campaign served partly as a training exercise for 2021 data collection, but also to provide extensive sampling over different crops and cropping systems within Ghana. Three teams surveyed three $\sim 50\,\text{km} \times 50\,\text{km}$ sites in the Guinea Savanna, Transition and Deciduous forest regions (see Fig. 1 for locations) between December 12 and December 21, 2020. The site locations and main crops associated to each site were chosen based on previous knowledge and in discussion with local agricultural extension workers. Within each site, crop types were initially identified from a drive-thru "windshield survey" (Defourny et al., 2014) and GPS locations and photographs were simultaneously gathered using mobile phones inside

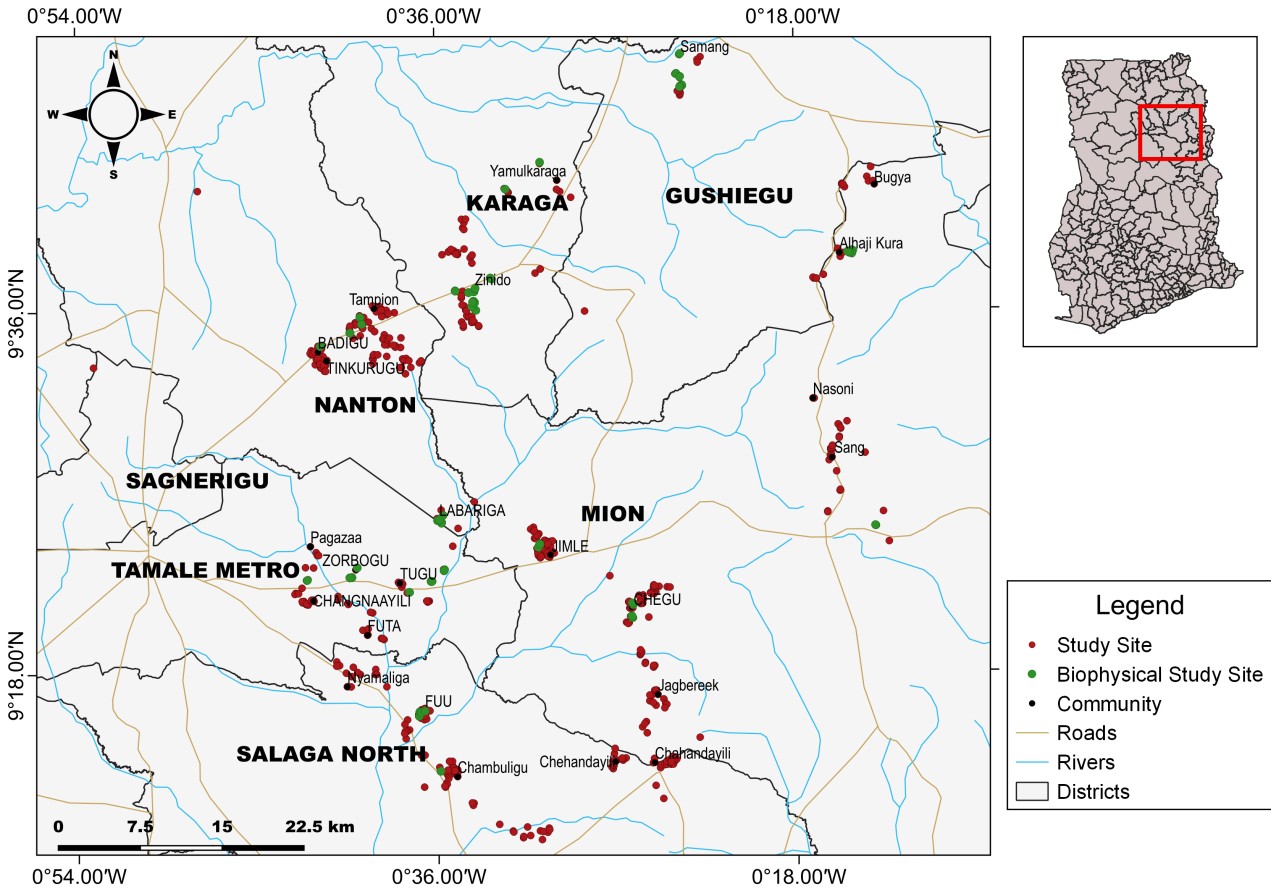

**Figure 2.** Location of the crop type mapping for the 2021 campaign. Red dots show agricultural fields. Green dots show locations where biophysical parameters were collected. Top right sub-image shows the location of the area within Ghana

individual fields. Crop type was mapped at a point location within each field. Fields that were less than $0.25\,\mathrm{ha}$ were ignored, following the recommendations from Defourny et al. (2014). Fields were selected for cases where: (i) a crop could be identified (the campaign took place in December when many fields had already been harvested, so crop identification was sometimes based on crop residue inspection); (ii) a single crop was present (so plots with evidence of intercropping were discarded). The crop types to map were selected as being the most representative of the region, with the goal of collecting around 600 points

for each region. The crop distribution per region is shown in Table 1. For the deciduous, savannah and transitional zone, 644, 630 and 660 fields respectively were sampled.

      A second crop type campaign was carried out in August 2021, more focused on identifying maize fields. It was conducted near Tamale (Guinea savannah region) (Fig. 2), with the primary aim of identifying maize fields to compare to the data collected in 2020, and a secondary aim of scoping fields that could be used for the biophysical parameter campaign. Fields were selected

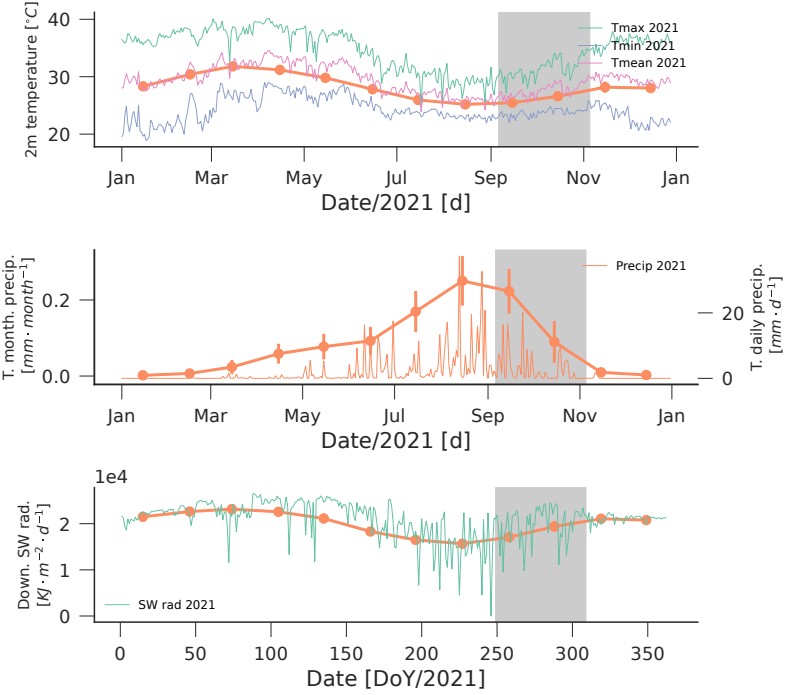

**Figure 3.** Monthly climatologies for temperature (top), precipitation and shortwave downwelling radiation over the Tamale area (derived from ERA5/Land dataset between 1990 and 2021), as well as daily temperatures, precipitation and downwelling radiation for 2021. Grey area indicates biophysical parameter collection period.

following the same considerations as the 2020 campaign, but rather than points, polygons of the field boundaries were collected. This required physical access to the fields, for which access permission was needed from the owners. This resulted in a smaller number of fields (375) being surveyed compared to the 2020 campaign (Table 1). Field boundaries were collected by walking around the field using a GPS tracker, and taking care to exclude trees and buildings to ensure that the mapped field area was as homogeneous as possible and measurements in that area would relate to the crop. Only minor post-collection editing was
performed (e.g. closing polygons).

### 2.3 Biophysical parameter collection

The collection of biophysical parameters in 2021 focused on maize farms around Tamale. These farms were selected from maize fields mapped in 2021 (Sect. 2.2) after permission was obtained from the farmers to allow repeated visits and harvesting at the end of the growing season. The field campaign was delayed by the arrival of the measurement equipment into Ghana and
training requirements (covid and related delays), so the fields that were selected were those that appeared to be less developed towards the start of the measurement campaign (August 2021). In the study area, maize is typically sown around June, so it is possible that the selected fields are sown later than what is usual for this area.



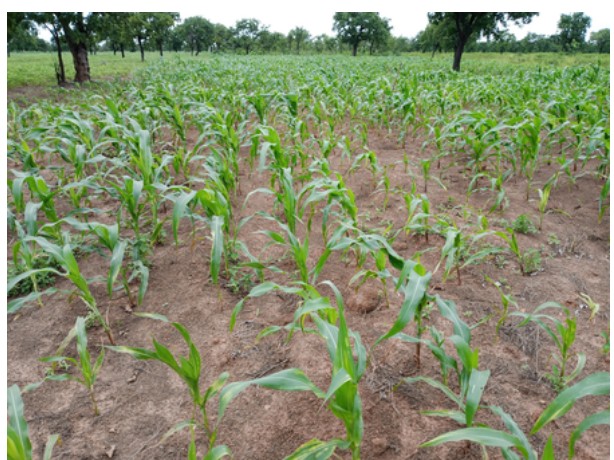 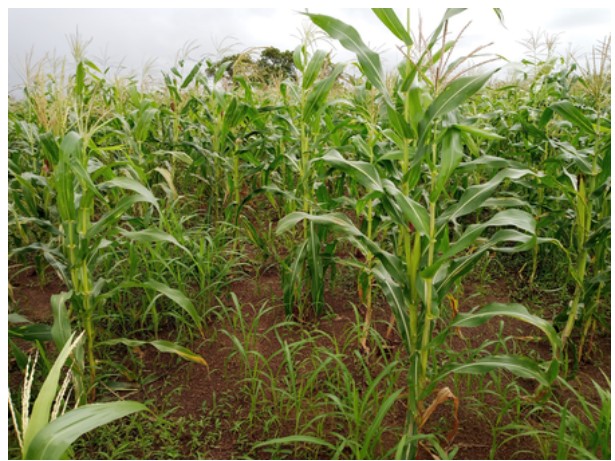

**Figure 4.** Pictures of two maize fields within the study area, showing the crop heterogeneity, present of weeds and trees within fields. (Left) field 7071ZIN (17th September 2021) *In situ* LAI: $1.1\,\mathrm{m^2/m^2}$, Chlorophyll conc: $39.3\,\mathrm{\mu g cm^{-2}}$). (Right) field 3075TAM (14th September 2021) LAI: $1.9\,\mathrm{m^2/m^2}$, Chlorophyll conc: $45.2\,\mathrm{\mu g cm^{-2}}$.

An initial set of 56 maize fields were selected both for biophysical parameter and yield characterisation. The field sizes ranged from $0.25\,\mathrm{ha}$ to $2.2\,\mathrm{ha}$, with an average field size of $0.78\,\mathrm{ha}$. From the initial 56 farms, eight farms were dropped later in the season as the crop had been damaged by animals, flooded or abandoned by the farmer and overgrown with weeds.

Measurements were taken from September 6th, 2021 to November 5th, 2021. Measurement of LAI and leaf chlorophyll content was performed weekly in the sample farms using an Li-Cor LAI-2200-C device and a Minolta SPAD 500 device respectively. The full field protocol for LAI and leaf chlorophyll is presented in Appendix A.

In each field visited, 4 locations were selected and marked, and measurements taken at these. For the leaf chlorophyll measurements, the 5th and 6th leaf (relative to the bottom of the canopy) of individual plants in each sampling site were tagged on the first visit. Measurements were then repeatedly taken on these leaves in subsequent visits. At the time of measurements, the general state of the crop and its phenology were also noted.

## 2.4 Crop yield and biomass measurements

Crop yield was measured by crop cutting. For each farm, three $6\,\mathrm{m} \times 6\,\mathrm{m}$ plots inside the field were harvested, cobs removed, and grain weighted. We report the quadrant yields, as well as the field-averaged mean yield and associated standard deviation. For a subset of 10 farms, above ground biomass was also sampled. The full protocol is given in Appendix B.

## 2.5 Satellite data

Together with the ground data described above, we have also produced an analysis-ready dataset (ARD) of contemporaneous satellite observations to facilitate training and experimentation for dissemination. We use the ground data to develop an empirical estimation of LAI. We have used the Planet Surface Reflectance (SR) version 2 product (Planet, 2018) to give sufficient



| 2020 campaign | | | | | | 2021 campaign | |
|---|---|---|---|---|---|---|---|
| **Deciduous** | | **Guinea Savannah** | | **Transition** | | **Guinea Savannah** | |
| **Crop class** | **Number** | **Crop class** | **Number** | **Crop class** | **Number** | **Crop class** | **Number** |
| Oil palm | 117 | Maize | 90 | Maize | 127 | Maize | 214 |
| Maize | 116 | Soybean | 87 | Cassava | 83 | Groundnut | 71 |
| Cocoa | 90 | Sorghum | 75 | Cashew | 78 | Cassava | 4 |
| Cassava | 74 | Rice | 74 | Yam | 60 | Rice | 50 |
| Plantain | 66 | Groundnut | 68 | Plantain | 54 | Soyabean | 20 |
| Orange | 61 | Pigeon pea | 56 | Cocoa | 53 | Cowpea | 5 |
| Rubber | 46 | Yams | 56 | Tomatoes | 53 | Yam | 6 |
| Rice | 45 | Millet | 52 | Mango | 41 | Millet | 1 |
| Coconut | 10 | Cassava | 39 | Garden eggs | 33 | | |
| Garden eggs | 7 | Cowpea | 30 | Pepper | 26 | | |
| Bush | 12 | Bush | 3 | Cabbage | 23 | | |
| | | | | Orange | 18 | | |
| | | | | Cowpea | 11 | | |

**Table 1.** Number of mapped fields per crop type class for the 2020 and 2021 campaigns.

temporal sampling over the crop season, as cloud cover prevalent during the rainy/growth season limits the use of other optical data such as from Sentinel-2 or Landsat for much of that time. The goal here is to demonstrate the application of the field data to calibrate a simple NDVI to LAI model to provide a spatial estimate of LAI for each sample field and extend the LAI dataset. We choose NDVI as the surrogate for mapping LAI it is a commonly used vegetation index that is frequently used to describe
crop condition and yield (Turner et al., 1999; Smith et al., 2002; le Maire et al., 2004; Ferwerda and Skidmore, 2007; Le Maire et al., 2008).

Surface reflectance data were subset and downloaded from Planet Explorer (https://www.planet.com/explorer/). This product is derived from the top of atmosphere (TOA) radiance images acquired by the PlanetScope constellation which collects data in the red, green, blue and near infrared bands with a nominal resolution of $\sim 3.7\,\mathrm{m}$. The SR product has a ground
sampling distance of $\sim 3\,\mathrm{m}$ and a positional accuracy better than $10\,\mathrm{m}$ (Planet, 2018). The data are atmospherically corrected and have an associated cloud, cloud shadow, etc. pixel mask (Planet, 2018). Even so, the vast changes in acquisition geometry, sensor properties, failure of the cloud and cloud/shadow mask and inconsistencies in the atmospheric correction result in the



measurements from Planet being very noisy and contaminated with outliers. Outliers and gaps in the time series (particularly at the start of the measurements period) require treatment: we develop here a robust smoothing and interpolation approach that

allows us to achieve the desired LAI mapping, along with an estimate of LAI uncertainty.

We use an efficient and robust smoothing filter with a bi-square weighting to flag and remove gross outliers in the Planet NDVI time series (Heiberger and Becker, 1992; Garcia, 2010). An outlier is flagged if $\left|\frac{u_i}{4.685}\right| \geq 1$, where $u_i$ is the studentised residual for sample $i$ (Garcia, 2010). An example application of the smoother is shown in Fig. 5(a). To further reduce the large remaining variability, we fit a double logistic function (Zhang et al., 2003; Beck et al., 2006; Atkinson et al., 2012; Yang et al.,

2019) (Eq. 1) to NDVI as a function of time $t$ for each pixel, an effective way to both reduce the noise (Hird and McDermid, 2009; Jönsson and Eklundh, 2002) and allow temporal interpolation.

$$\text{NDVI} = p_0 - p_1 \left[ \frac{1}{1 + \exp(p_2(t - p_3))} + \frac{1}{1 + \exp(-p_4(t - p_5))} - 1 \right] \tag{1}$$

We have implemented the double logistic fitting as a two stage process to account for both the large variability and the limited observational opportunity at the early start of the growing season. As a first stage, the six parameters $p_i$, $i \in (0...5)$ in

Eq. 1 are estimated for individual pixels over a field. Then, the per field median values for the six parameters are calculated and used to define bounds in parameter values for the second pass. In the second pass, the double logistic is again fitted the per pixel NDVI time series, with all parameters except $p_1$ (the amplitude) being constrained to be within $5\,\%$ of the median field value. In this way, we ensure that the mapped timing information is spatially correlated at the field level, and that the variation in the amplitude of the vegetation index will be greater.

The processing described above results in more stable estimates of NDVI over time, as can be seen in Fig. 5(c), particularly tightening up the temporal trajectory towards the start of the time series. We use this interpolated and smoothed NDVI data to develop the mapping to LAI. A potential issue with a mapping from NDVI to LAI are saturation effects with high (Baret and Guyot, 1991). For maize in the study area, very high LAI is never achieved, and the field measurements never exceed an LAI of 3, so we might suppose that saturation of the signal should not be a problem here. The limited range of the field data LAI

data also suggests that a linear model is an acceptable model choice. We estimate the value of NDVI on the day of the *in situ* observations from the smoothed/interpolated Planet data, and average both the EO estimated NDVI and the *in situ* LAI over the field. We randomly split the data set set into $70\,\%$ for training and $30\,\%$ validation. We fit the linear model $\text{LAI} = m \cdot \text{NDVI} + c$ to the training data and test its performance on the validation samples.

## 2.6 Validation of cropland masks

We use the collected data to partially validate the DigitalEarth Africa cropland mask (Burton et al., 2022). This binary (crop/no crop) mask has been developed for 2019, but there are plans to extend it to other years. Using crop masks from prior years is a pragmatic choice to monitor the same region in the current season (Becker-Reshef et al., 2018). We can use the crop location data to assess the accuracy of the crop mask for other years, and to test its suitability to enable within-season crop mapping. The DigitalEarth Africa cropland mask is based on a random forest (RF) classifier, but as well as binary mask of cropland/non-

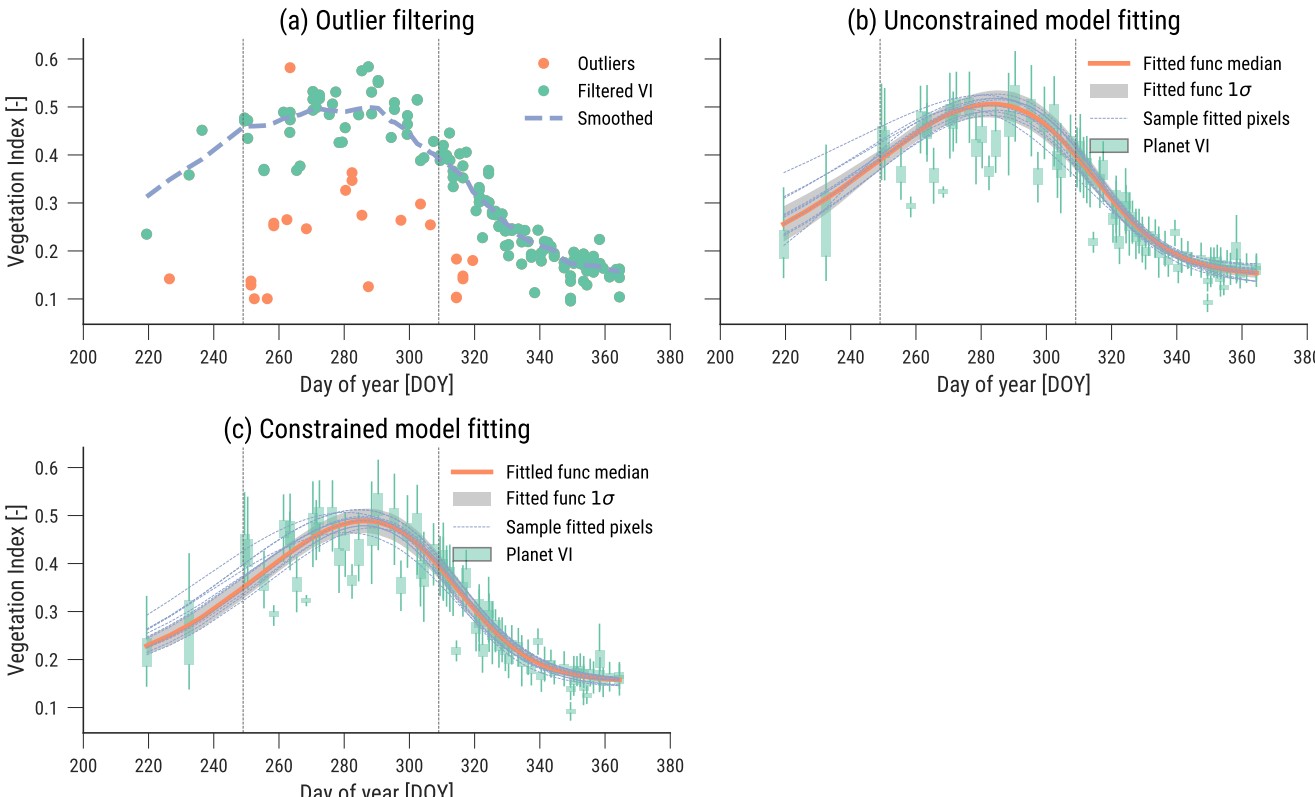

**Figure 5.** Planet VI time series processing steps example for field 7074ZIN. (a) Outlier filtering. (b) First pass single pixel double logistic fitting (unconstrained). (c) Second pass single pixel double logistic fitting (phenology parameters constrained by median field). The vertical lines show the extent of the ground campaign period.

cropland final product, it provides an estimate of the probability of each 10 m pixel being cropland. The final binary mask is derived from the pixel probability data set, and cropland is assumed if the pixel probability is greater than 0.5. We can assess the quality of the mask by testing the fraction of surveyed cropland locations (pixels) and their associated probability. A good mask would be characterised by a large proportion of the visited cropland pixels having a probability larger than 0.5. Conversely, a poor mask would show most visited pixels having a probability lower than 0.5.

**2.7    Crop mask classification**

The data sets collected in 2020 and 2021 can be used to extend previous efforts to provide cropland/crop type maps (Xiong et al., 2017; Jolivot et al., 2021; Estes et al., 2021; Burton et al., 2022), but here we illustrate the use of the 2021 data set in developing a 10 m maize mask for the whole Northern province in Ghana using data from Sentinel 2. The classification experiment is done using the Google Earth Engine platform (Gorelick et al., 2017). The approach taken was to collect Sentinel 225    2 observations of surface reflectance (atmospherically corrected with the sen2cor package (Louis et al., 2016), GEE data set





"COPERNICUS/S2_SR") between May and October 2021, when rain-fed crops are being grown. After applying a cloud mask, and only processing pixels labelled as "Crops" in the ESRI Sentinel 2 landcover map (Karra et al., 2021), temporal series of number of vegetation indices (NDVI, LSWI, IRECI and GCVI) and a subset of spectral bands (Red Edge 1, NIR, SWIR1 and SWIR2) are then smoothed/interpolated using a robust Whittaker smoother (Eilers, 2003; Garcia, 2010), with a smoothing

strength parameter of 0.5. The classifier used was a random forest (RF) with 100 trees, and in order to train the classifier, the individual pixels underlying the field-surveyed polygons were used. The pixels were split into two sets: 70 and 30 % of the points were used for training and validation (respectively). For the production of the final mask, all pixels were used to train the classifier.

## 3   Results

### 235   3.1   Biophysical parameter measurements

Pictures from two typical fields are shown in Fig. 4, which show the clear row structure and the low plant density that was common to most fields. Time series of the evolution of *in situ* measurements are shown in Figs. 6 and 7 for LAI and chlorophyll respectively for the sample fields.

LAI values ranged from 0.1 to $5.24\,\mathrm{m^2m^{-2}}$, with a mean value of 1.37. The 10, 50 and 90-th percentiles were 0.5281, 1.15

and $2.13\,\mathrm{m^2m^{-2}}$, respectively. LAI values are lower compared to other regions where irrigation and fertilisation are common, but are in line with other studies for the area (Srivastava et al., 2016; MacCarthy et al., 2015). In some of the fields the decrease of LAI from around its maximum is obvious (e.g., fields labelled as 1029ZIN, 5034TUG, 5036TUG, 1056ZIN). The pattern for other fields is not clear, with some having the LAI peak towards day of year (DoY) 275 (e.g., fields labelled as 7021YAM, 7068ZIN, 7069ZIN), whereas other fields show no clear dynamics.

For leaf chlorophyll concentration, the values ranged between 6.1 and $60.3\,\mathrm{\mu g cm^{-2}}$, with a mean value of $34.2\,\mathrm{\mu g cm^{-2}}$ and the 10, 50 and 90-th percentiles given by $15.71\,\mathrm{\mu g cm^{-2}}$, $35.9\,\mathrm{\mu g cm^{-2}}$ and $49.39\,\mathrm{\mu g cm^{-2}}$.

The trends for leaf chlorophyll concentration are clearer than those of LAI. Most fields show an expected decay of chlorophyll as the season progresses, with most of the differences between fields relating to the timing and magnitude of the leaf chlorophyll reduction.

### 250   3.2   Crop yield and biomass measurements

The distributions of measured yield are shown in Fig. 8a. Biomass and harvest index histograms are shown in Fig. 8b. For the individual quadrant measurements, yield varied between $35\,\mathrm{kg\,ha^{-1}}$ and $5036\,\mathrm{kg\,ha^{-1}}$. The per field averaged values were between $190\,\mathrm{kg\,ha^{-1}}$ (field 7033FUU) and $4580\,\mathrm{kg\,ha^{-1}}$ (field 7021YAM), with an average of $1379\,\mathrm{kg\,ha^{-1}}$ and a standard deviation of $872\,\mathrm{kg\,ha^{-1}}$. The uncertainties within the fields were also important, with an average within-field standard

deviation deviation in yield $\sim350\,\mathrm{kg\,ha^{-1}}$. Total above ground biomass was between 1000 and $16\,000\,\mathrm{kg\,ha^{-1}}$, and the harvest index varied between 0.21 and 0.77.

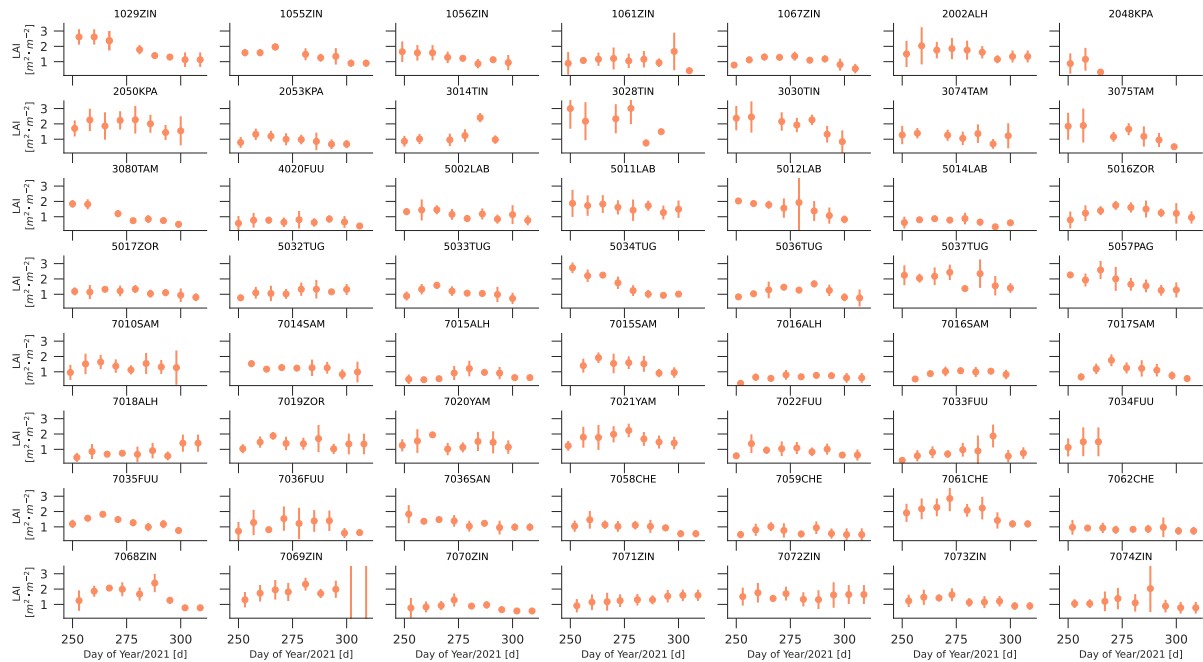

**Figure 6.** Per field temporal evolution of *in situ* leaf area index (LAI). For any date, the mean is shown, and error bars represent $\pm 1\sigma$.

### 3.3 EO-derived leaf area index (LAI)

The approach described in Sect 2.5 results in a simple transformation between Planet NDVI and LAI. The calibration and validation of this approach are shown in Fig. 9. The model performs in line with medium resolution products (Fang et al., 2019), with a validation root mean squared error (RMSE) around $0.44\,\mathrm{m^2m^{-2}}$, and a negligible bias (Fig. 9). Fig. 9 clearly shows an underestimation of the Planet NDVI signal for $LAI > 1.5$. This effect is likely to be caused by the averaging of field measurements to field scale. Other possible explanations may be the due to the impact of the soil in the NDVI signal (Carlson and Ripley, 1997). A comparison of the field LAI measurements and the Planet-derived LAI time series is presented in Fig. 10.

### 3.4 Validation of cropland masks

Fig. 11 shows the distribution of DEAfrica's cropland mask probabilities for the surveyed pixels in 2020 and 2021. For a perfect mapping, one would expect the cumulative distribution to be a Heaviside step function changing from 0 to 1 for a high probability value, suggesting that all the pixels are detected as cropland with a high confidence. At the very least, the change point should be around 50. Any samples that appear with less than $50\,\%$ probability would be omission errors.

There are clear differences between years and sites. For the semi-deciduous zone in 2020, the vast majority of pixels are labelled as non-crop, with the cropland mask consistently under-reporting crop area. Best results are obtained for maize, cassava and plantain, where $\approx 12\,\%$ of crop area is reported as such. For the transition zone in 2020, the performance is better.

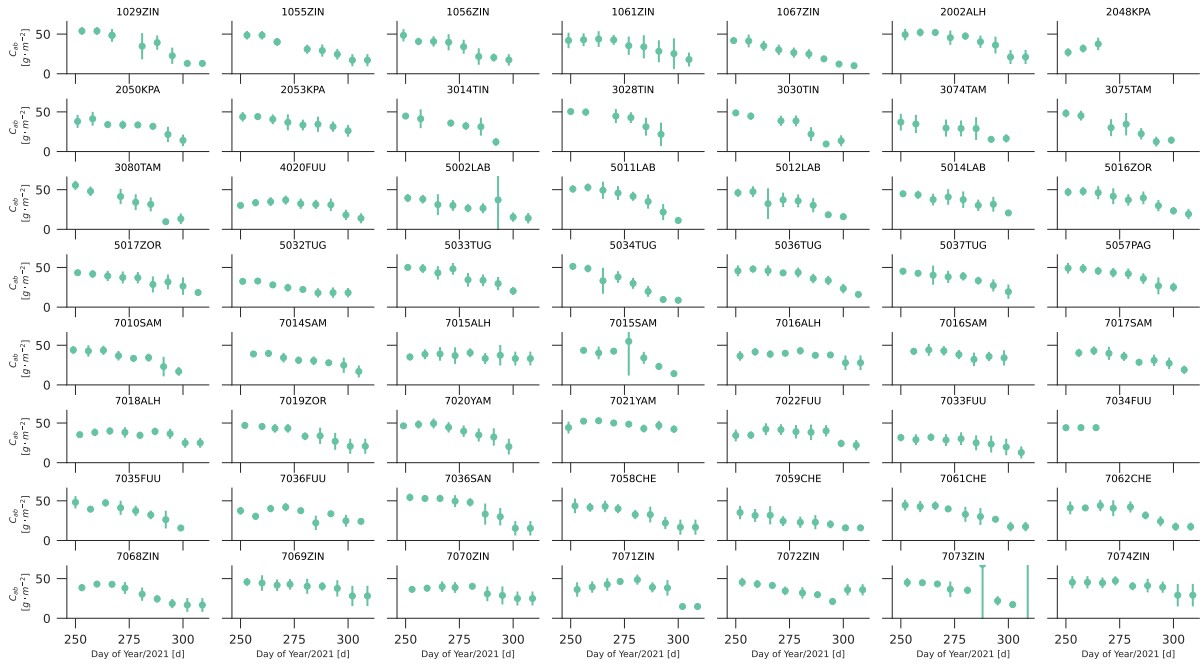

**Figure 7.** Per field temporal evolution of *in situ* leaf area chlorophyll concentration ($C_{ab}$). For any date, the mean is shown, and errorbars represent $\pm 1\sigma$.

For pepper, cowpea and cabbage, $\approx 50\,\%$ of the surveyed pixels are reported as cropland. For maize in this region, only $40\,\%$ of pixels have a probability of more than 0.5. Results for the savanna region 2020 are better: $\approx 60\,\%$ of maize fields are labelled as cropland, but for other popular crops such as sorghum, millet, groundnut and soybean, less than $40\,\%$ of the samples are labelled as cropland. For the savanna region in 2021, results are similar to the same region 2020, with rice and maize having around half of the pixels detected as cropland by the mask. Soybean is only detected in $20\,\%$ of pixels. The conditions towards the South of Ghana (persistent cloud, mixing of crops and trees) make it harder to identify cropland. Towards the North, with fewer trees and (comparatively) less cloud, the cropland mask improves, but still misses around half of the cropland area, and for some crops (soyabean), the situation is even worse.

It is worth noting that for 2020, each pixel belongs to a different field, whereas for 2021, all pixels belonging to the visited fields have been considered, and as such, pixels within a field are spatially correlated, which suggests that the result for this site is probably optimistic. An example of the cropland mask and the visited fields around a small area within the Mion district is shown in Fig. 12, where it is clear that the cropland mask misses many of the visited fields. The reported probability of cropland values are also fairly low (very few pixels have probabilities over $70\,\%$), suggesting a large uncertainty for most pixels.

Our results are hard to compare with the the official validation report from (Burton et al., 2022), as we are considering using the mask for a different periods than those used to develop the data set, but in our case, the results show a large omission error

(a) Crop yield.

(b) Above ground biomass, harvest index.

**Figure 8.** Crop yield distribution (top left: field averaged, bottom left: individual quadrants), field averaged above ground biomass (top right) and field averaged harvest index (bottom right).

of more than $50\%$ for all crops, with results being markedly worse towards the South of Ghana. These figures is at least double the the reported $25\%$ omission reported in Burton et al. (2022).

### 3.5 Crop type classification

The maize mask derived as presented in Sect 2.7 is shown in Fig. 13. The validation results (using the 70:30 data split introduced earlier) were an overall accuracy of 0.84 and a kappa score of 0.68 (see Table 2 for details).

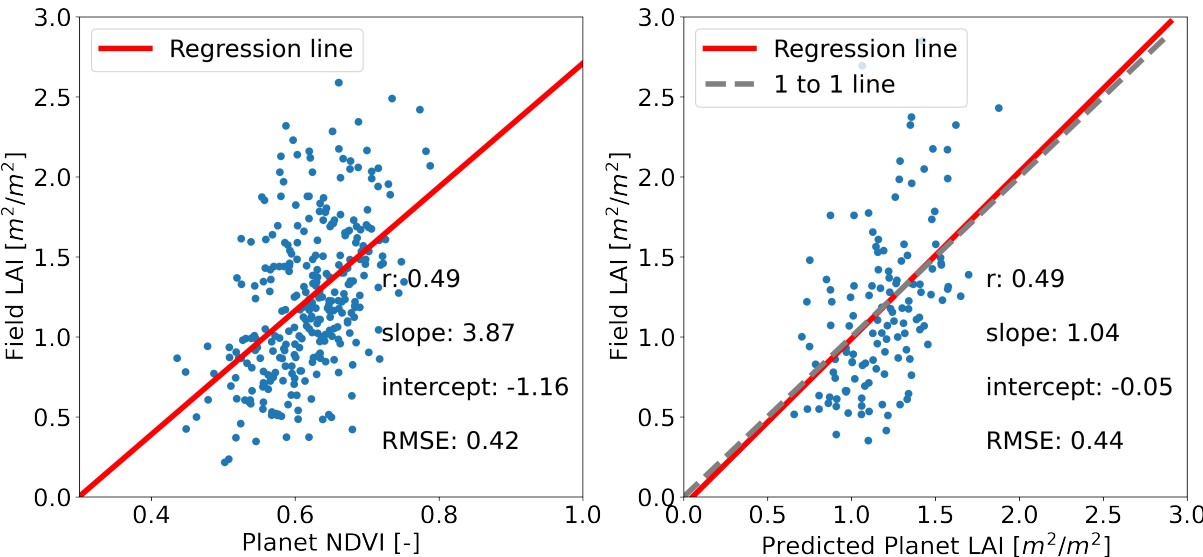

**Figure 9.** NDVI to LAI calibration (left) and validation (right)

| Crop | Other | Maize | User's accuracy | Overall accuracy | Kappa score |
|---|---|---|---|---|---|
| Other | 7823 | 1234 | 0.86 | | |
| Maize | 1800 | 7975 | 0.82 | 0.84 | 0.68 |
| Overall accuracy | 0.81 | 0.87 | | | |

**Table 2.** Results from the crop classification for 2021.

## 3.6 Yield prediction testing

The relationship between leaf area and crop yield has been widely explored, often via vegetation indices (e.g., Becker-Reshef et al. (2010)), and it usually follows that a green, healthy and dense canopy results in higher yield. Many authors relate yield
to the magnitude of the maximum value of leaf area (or vegetation) index. This is a fairly crude relationship, but for regional applications in areas with large fields and monocultures, it can be quite effective (Mkhabela et al., 2011; Petersen, 2018; Kouadio et al., 2012). Here, we look at this relationship between maximum field measured LAI and field measured yield, as well Planet-derived LAI and field measured yield. For *in situ* data, a linear relationship yields a reasonable fit (coefficient of correlation $R = 0.66$, coefficient of determination $R^2 = 0.44$), once fields 7021YAN, 7033FUU and 7036SAN are removed
as outliers. The slope and intercept of the linear model are 866.44 and -180.45, respectively (Fig. 14a). Including the left out fields reduces the coefficient of determination to $R^2 = 0.31$. Using the maximum $LAI$ value from the Planet data introduced in Section 2.5, the correlation degrades to $R = 0.53$ ($R^2 = 0.29$, ignoring fields 7021YAN, 7033FUU and 7036SAN) (Fig. 14b). Slope and intercept for the Planet derived maximum $LAI$ are 1606.51 and -893.10 respectively. The lower correlation from





**Figure 10.** Predicted LAI time series and associated standard deviation (blue line and grey area) in comparison with field measurements (per field mean and standard deviation, green lines).

the satellite-derived relationship is probably explained by the underestimation of the maximum LAI for the higher yield fields



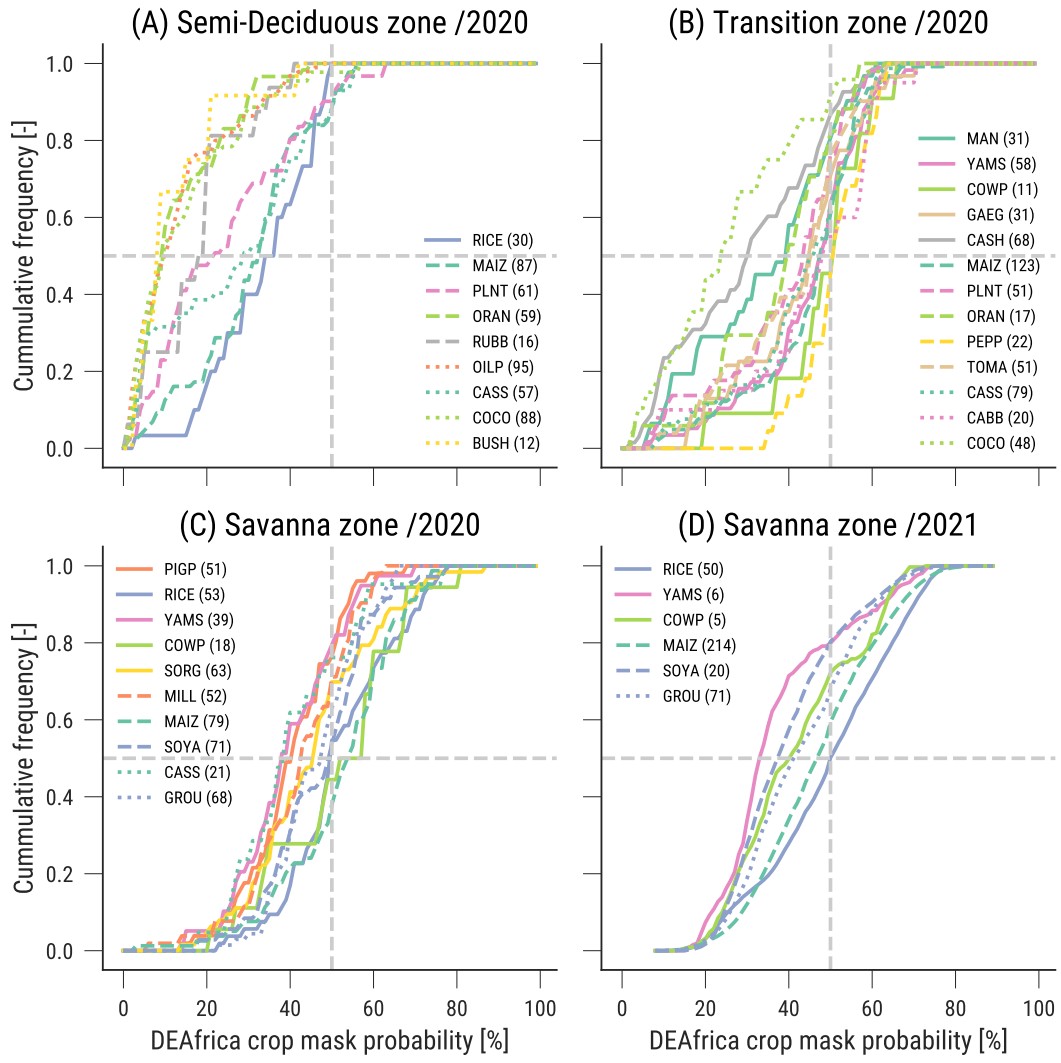

**Figure 11.** Cumulative frequency of surveyed pixels against probability of cropland per DEAfrica cropland probability mask for 2019. Horizontal dashed lines are 0.5 and vertical line is cropland probability of $50\%$. Numbers between brackets in legends indicate the number of samples of each class included in the calculations.

(Fig. 9). For the *in situ* case, the value of the slope in the maximum $LAI$ relationship is close to $900\,\mathrm{kg\,ha^{-1}}$, suggesting that the with errors in the Planet $LAI$ estimates around $0.44\,\mathrm{m^2 m^{-2}}$, the typical error in the estimate would be around $381\,\mathrm{kg\,ha^{-1}}$, a value close to the average uncertainty in the field measured yield.

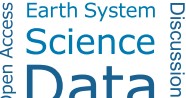

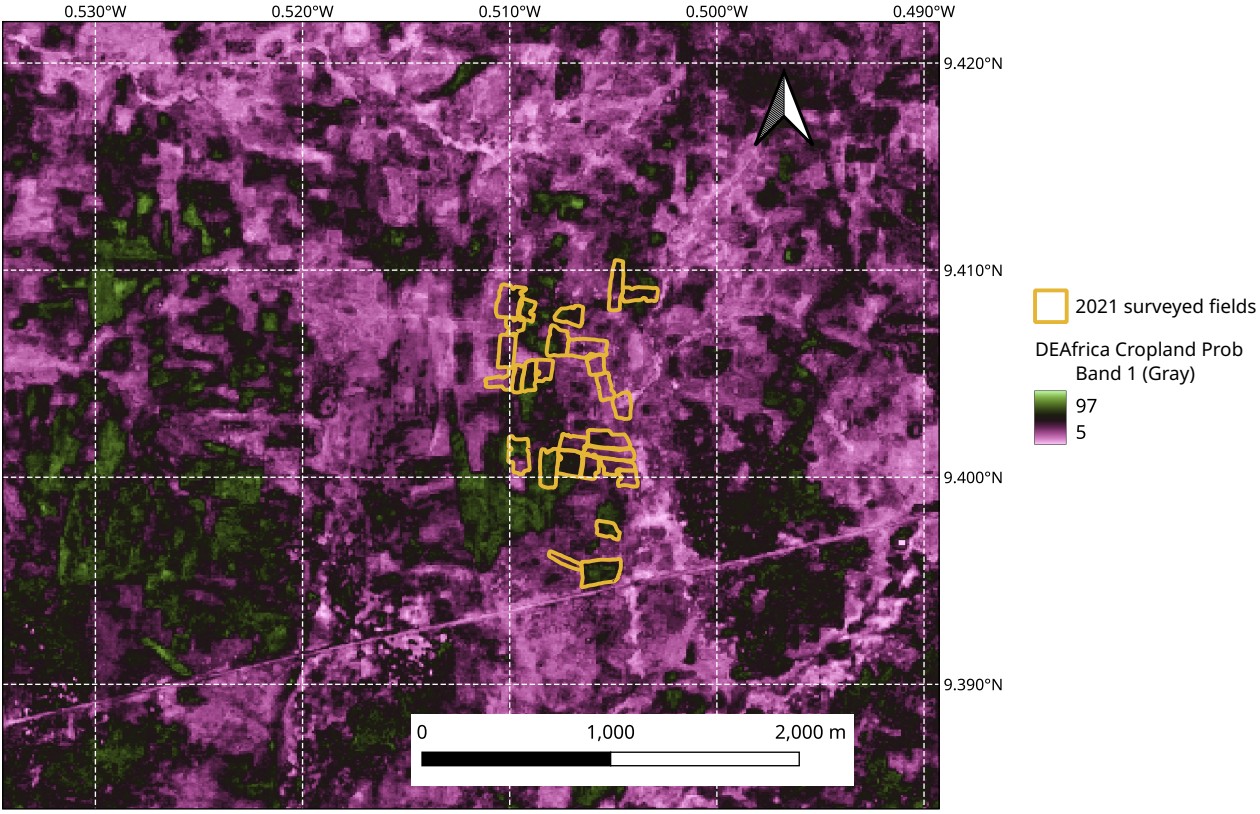

**Figure 12.** Detail of some the visited field sites (yellow polygons) and the DEAfrica cropland probability mask (Burton et al., 2022) around Mion districts in the Northern region in 2021.

## 4 Discussion

### 4.1 Crop measurements

The *in situ* data gathered is a useful contribution to advance crop monitoring methods for smallholder farmers in West Africa. It is important to note that the data collection started when the crops were already established in the fields, as is clear from the satellite temporal trajectories (e.g. Fig. 5 and Fig. 4), so that the initial growth period after emergence was not captured. Additionally, due to logistical issues, the field campaign could only start towards the end of August/start of September 2021, when planting in this area starts three months earlier. To capture the longest possible dynamics over the campaign, the chosen
fields were all late-sown, which may result in them not being very representative of the majority of fields in the region. The choice of fields with the least amount of tree cover and no intercropping may also impact the generality of the data gathered

**Figure 13.** (Left panel) Maize mask (maize shown in orange). Insets (A) and (B) illustrate the field data. Maize fields are shown as green outlines and other crops as red outlines .

with respect to other fields where these conditions are not met. The *in situ* LAI measurements show some variability (mean standard deviation for samples collected on the same date on the same field of $\approx 0.25\,\mathrm{m^2 m^{-2}}$). The effect of this variation is relatively more important here as the maximum LAI is quite small ($\approx 2\,\mathrm{m^2 m^{-2}}$, compared to e.g. US, where the maximum

LAI will be between around 5-6 $\mathrm{m^2 m^{-2}}$ (Nguy-Robertson et al., 2012).

The variation of leaf chlorophyll content appears different to the LAI trajectory, but is broadly characterised by a plateau followed by a steady drop. Here, we have not looked at its use, either for EO-product validation or other applications, as the scarcity of Sentinel 2 reflectance data over the fields makes retrievals unreliable.

There is a large variability of yields within individual fields. This is due to differential management practices, small-scale

soil and/or topography variations, etc. Three crop cuts over a standardised area were used to determine yield, conforming to common practice. However, the spatial locations of these estimates were not surveyed, so the data limits us answering whether




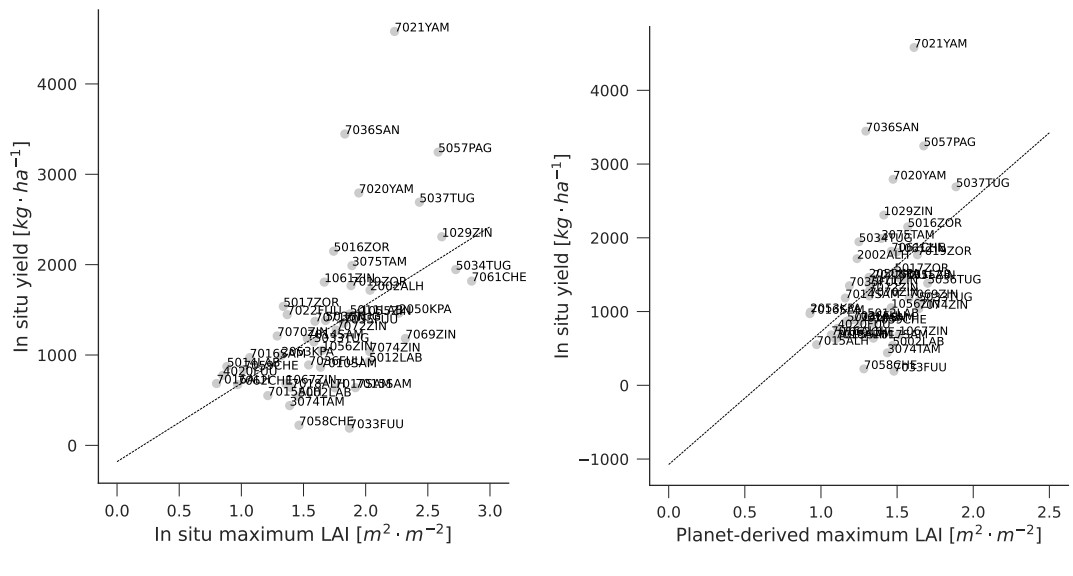

(a) *In situ* maximum LAI vs *in situ* yield.          (b) Planet-derived maximum LAI vs *in situ* yield.

**Figure 14.** Relationship between maximum LAI estimates and yield. For *in situ* measured LAI vs yield (Fig. 14a), and discarding fields 7021YAN, 7033FUU and 7036SAN, the linear fit has an $R^2 = 0.44$ and the linear fit is shown as the dotted line $y = 866.33 \cdot x - 180.45$.. For Planet-derived LAI vs yield (Fig. 14b), and discarding fields 7021YAN, 7033FUU and 7036SAN, the linear fit has an $R^2 = 0.28$ and the linear fit is shown as the dotted line $y = 1606.51 \cdot x - 893.10$.

EO data provides useful within-field yield variability. Yield estimates should also be co-located with the LAI/leaf chlorophyll measurement points to fully exploit the connection between canopy variables and yield.

Only limited data on above ground biomass was collected, but the distribution of the harvest index between 0.2 and 0.7
suggests that the relationship between biomass and yield is not well established. Lambert et al. (2018) finds a strong relationship between AGB and yield for maize in Mali. Karlson et al. (2020) shows that for a range of cereals, the relationship varies from year to year in Burkina Faso. Hay and Gilbert (2001) shows a HI ranging from 0.19 to 0.53 for maize in Malawi. It is interesting that Moser et al. (2006) point out that water stress can have an impact on HI in maize, which suggests that the relationship between AGB and yield may not be constant. This has important implications, as in many applications of EO, the target variable
is AGB which is then converted into yield assuming a fixed HI (Jin et al., 2017; Lambert et al., 2018).

### 4.2 LAI estimations

We presented a very simple approach to exploit the ground data set and calibrate a simple LAI relationship to satellite NDVI. The outlier filtering and logistic function fitting are fairly standard approaches, and are necessary pre-processing stages to establish a linkage between ground and satellite measurements. While the results of the mapping appear in line with similar
approaches, and better than universal relationships for maize (Kang et al. (2016) report an RMSE $\approx 1 \, \mathrm{m^2 m^{-2}}$ for maize), it is





clear that there is an issue with high LAI samples in Fig. 9. Given the low magnitude of LAI in the problem, it is unlikely this is a saturation effect, but rather a limitation of not having accurately co-located EO and field measurements.

## 4.3 Cropland mask validation

We have used the crop location data to assess a continent-wide cropland mask and to derive a local maize mask. Both were
developed with Sentinel 2 data, which in Ghana (and similar tropical regions) is challenging due to limited observation opportunity due to clouds. In Ghana, this is exacerbated as the vast majority of crops are grown during the wet season. Additionally, smallholder landscapes show a large within-class variability, due to large differences in management choices, as well as the presence of trees and other non-crop vegetation. So infrequent temporal sampling and most crops growing simultaneously makes using Sentinel 2 challenging for developing a cropland mask. The task of developing a crop type mask is even harder,
and in all likelihood, would need vast numbers of crop labels covering different seasons and locations.

The Burton et al. (2022) cropland mask was assessed for the two subsequent years to that when it was produced. We found that in areas towards the South of Ghana (semi-deciduous agroecological zones), the cropland mask underestimates cropland area for most crops. Towards the transition and northern Savanna zones, the performance improves, but in the best case, omission errors are larger than $50\%$. This is in marked contrast to the validation report for West Africa, where omission
errors are $50\%$. It would be of great interest to repeat the exercise here for the cropland masks for the years 2020 and 2021, and assert whether the performance is similar (suggesting that the issue is with the data source and method) or whether they improve, suggesting a very dynamic interannual cropland variation. As more of these data sets become available (e.g. Estes et al. (2021)), the data we provided in this contribution can be used as a source of independent validation, but also as a source of training data, with the important caveat that no non-cropland samples were collected. Non-cropland ssamples are needed to
fully characterise the masks.

The maize mask that was developed in this paper demonstrates that the data can be used as an input to a classifier. However, the limited number of samples for 2021 (where the main aim of the field campaign was biophysical parameter collection) result in a crop mask that is probably only reliable around the data points. Also, since the surveyed fields were selected as late sown, this may also bias the field selection. We note that even methods based on more training data (>4000 samples), more complex
classifiers and a very rich set of data combining Sentinel 1, Sentinel 2 and Planet (M Rustowicz et al., 2019) still report overall accuracies for crops in Ghana around $60\%$.

It is instructive to compare the results for Ghana to those in Nigeria. In Nigeria, cropland and crop type masks appear feasible with accuracies over $70\%$ for both cropland and crop type reporteb y Ibrahim et al. (2021) and even over $90\%$ (using Sentinel 1 and 2) (Abubakar et al., 2020). These two studies suggest that if cloud cover is not an issue, Sentinel 2 is able to use the
temporal signal to map different crops, and adding Sentinel 1 (as in Abubakar et al. (2020)) only marginally improves on the Sentinel 2 results. The importance of the optical data suggests that there might be limited improvements in crop type mapping using Sentinel 1 for Ghana.



## 4.4 Yield prediction

We show the effect of using maximum LAI to predict yield, both using the *in situ* and satellite-derived LAI estimates at field
level. After removing three outlier fields, we find that the relationships are quite weak ($R^2 = 0.44$ and $R^2 = 0.28$ for ground
and EO-derived maximum LAI, respectively). The poor performance of the EO-derived method stems from the saturation
effect, which strongly affects the maximum LAI estimation. For the ground data, the results still show a large dispersion.
The slope of the regression is also large: $866.33 \, \mathrm{kg \, ha^{-1}}$ per unit of LAI. Any error in maximum LAI estimation will have a
considerable error in yield estimation. Nevertheless, the results reported here compare favourably to more complex field-level
studies done in the US corn belt ($R^2 = 0.45$, RMSE $1850 \, \mathrm{kg \, ha^{-1}}$) using a more complex model and Landsat data (Deines
et al., 2021). Kang and Özdoğan (2019) use a data assimilation system ingesting MODIS and Landsat data over the US corn
belt and report yield estimates with a correlation $R^2 = 0.41$ and RMSE $2170 \, \mathrm{kg \, ha^{-1}}$.

The above discussion hints some of the challenges of monitoring yield in smallholder landscapes: the large within and
between field variations in yield only have a modest effect in LAI. Given that these variations occur over small areas with
roughly the same meteo drivers, the variability of the system can only be studied by making use of wall-to-wall observations
of e.g., LAI, as other sources of information (e.g., agrometeorological crop models) will not exhibit variation in its drivers over
these spatial scales.

Our results suggest that extracting relationships between yield and EO-derived diagnostics such as maximum LAI is uncer-
tain in part due to yield showing a large spread even within single fields, indicating that the scale of analysis should be the
point within the field, rather than the field average. With precisely co-located yield and LAI ground measurements, a clearer
understanding of the uncertainty of the canopy variable to LAI mapping could be developed.

## 5 Conclusions

We have gathered a rich data set to understand and develop crop monitoring methods using EO for maize in northern Ghana.
The data set includes the location of crops, a comprehensive set of repeated biophysical parameters (LAI and leaf chlorophyll
concentration) over the growing season, and measurements of crop yield and biomass. These measurements were acquired in
2021 (with some crop locations also reported in 2020), and were taken from an agricultural area East of the city of Tamale.
The collected data is novel in that it focuses on smallholder maize farms, an important and understudied agricultural landscape
that supports many farmers in Africa.

This data set has a number of uses, some of which we illustrate throughout this paper. The crop location data presented here
complements recent contributions such as Jolivot et al. (2021), and can also be used to validate and create cropland/crop type
masks. We demonstrate both of these uses in the paper. In West Africa, producing these data sets faces important challenges
due to similar timing of crop development, persistent cloud cover and huge heterogeneity in crop development within and
between growing seasons. This points out for a need for more multi-location, multi-season *in situ* data, to which this paper is
a small contribution.





The *in situ* collected biophysical parameters have been used to develop and validate a simple method to infer LAI from Planet data, which resulted in a local estimates having a low error (RMSE $0.44\,\mathrm{m^2m^{-2}}$, negligible bias), but a low coefficient of correlation $R = 0.49$, due to an underestimation of high LAI. The measurements described in this contribution can also be used to validate biophysical parameters retrieved using other sensors and methods (Fang et al., 2019; Delegido et al., 2011; Clevers and Gitelson, 2013; Brown et al., 2021; Kganyago et al., 2020). The repeated temporal sampling is an important feature

of this data set: as more sophisticated techniques are developed to fully use time series of data (Lewis et al., 2012), repeated measurements covering the entire development of the crop should be preferred over the traditional approach of data collected over a single or few dates. The multi-parameter nature of the data (LAI and chlorophyll concentration) is important in showing the dynamics of both and supporting efforts to define a rich set of Essential Agricultural Variables (Whitcraft et al., 2019).

    The data show a large variation in yield within a relatively small area and a considerable yield variation within fields. This

local variability cannot be attributed to coarse scale weather patterns or broad soil classes, but rather to local soils variation, farming practices, etc. There are important implications for the use of crop growth models in these systems and the need to consider the sources of the underlying yield variability, in addition to usual climate and soil drivers (Beveridge et al., 2018).

    The large yield variability also suggests that to truly understand food production, the real spatial distribution of yields needs to be measured. This will require the use of EO-based methods, and an important data collection effort similar to the one

presented here to understand their limitations. We show that there is a relationship between the *in situ* measured maximum LAI value and yield at the plot level, but the relationship deteriorates for the EO-derived maximum LAI, due to the additional uncertainty and bias of the EO-derived LAI estimate. Developing robust methods to infer LAI from EO data (in particular, high revisit frequency sensors) is critical to be able to monitor crop development.

    Providing cropland and crop type maps for smallholder systems is challenging. We use our data to test a cropland mask,

and find it underestimates cropland. We also use our data to demonstrate a local maize/non-maize mask. In the light of the limitations shown in both of these efforts, the collection of more extensive multi-site, multi-season crop location ground data is critical, as is the exploration of very dense time series, as in many tropical sites with dry/rainy season dynamics, most of the crop growing happens during the wet season, resulting in similar temporal dynamics and low opportunity of observation.

    The data set has some limitations: it covers a single growing season over a small area. Similar efforts for other years and

areas would greatly strengthen any study that makes use of this data. Second, field measurements were done preferentially in fields that were sown late in the season, and broader sampling of earlier and late-sown crops would be beneficial for an area where sowing occurs over a long period in the rainy season. Finally, while the data set allows us to consider yield as a function of LAI and/or chlorophyll, it doesn't allow us to understand what factors had a role in crop development, such as e.g. soil, fertiliser, management, etc. Finally, yield measurements within a field were not georeferenced, and this makes it hard to use

EO data to understand the important within-field variability.

    In the light of this, we suggest that campaigns looking at agricultural applications, should consider repeated measurements over the growing season, as well as considering multiple seasons or multiple geographical regions. In addition, measurements of crop yield should also be made to coincide with biophysical parameters, as well as detailed management information, as this would allow understanding the variability in yield estimates even within and between fields.

Collecting the data outlined above is expensive and challenging in smallholder agricultural landscapes, and in many cases, it will require a close conversation and collaboration with the farmer. Providing useful satellite-derived data products for e.g., agricultural extension workers, who work closely with the farmers might encourage the collection and sharing of more data like that shown in this paper.

# 6   Code availability

Code for the classification introduced in Sect. 2.7 is available from https://code.earthengine.google.com/4795796bb1f47ff7e9 ca9c0aae263c11

# 7   Data availability

The following data is available from http://doi.org/10.5281/zenodo.6632083 (Gomez-Dans et al., 2022):

**Detailed field sampling locations**  A GeoJSON file with the locations of maize fields where detailed measurements of bio-
physical parameters were made.

**Field location campaigns**  Four GeoJSON files with the locations of fields and the crop type.

   **Transition/2020**  Points located inside fields.

   **Deciduous/2020**  Points located inside fields.

   **Savannah/2020**  Points located inside fields.

**Savannah/2021**  Field outline polygons (avoiding trees).

**In situ biophysical parameter time series**  Time series of repeated measurements of leaf area index and leaf chlorophyll con-
       centration, as well as phenology observations and general comments. CSV format.

**Crop yield and biomass**  Crop yield measured in three quadrants per field, and biomass measurements on reduced set of 10
       fields. CSV format.

**Planet-derived time series of LAI**  Per-field GeoTIFF files that have been derived from the original Planet data.

**Maize mask**  A maize/no-maize mask for 2021 as described in Sect. 2.7 in GeoTIFF format.

# Appendix A:  LAI and leaf chlorophyll measurement protocol

## A1   Leaf Area Index measurements

   1. Identify 4 locations in each of the fields to take the LAI measurements. Ensure the selected locations capture the vari-
ability in plant stand on the field.



2. Identify the rows in which measurements will be made and mark the locations with pegs for subsequent measurements.

3. At each location, take one above canopy reading and then 4 below canopy readings diagonally as illustrated in the picture below.

4. Ensure that the direction of the view cap is consistent for all readings per location.

5. Follow the steps in using the LI-COR 2200C in measuring LAI to capture the data.

Important:

  – The 4A measurements are necessary to deal with scattering corrections (generating $K$ records) if the sun is out. Otherwise one A reading and several B readings are enough.

  – Increasing number of below canopy (B) readings improves spatial average.

– In a canopy that is $1\,\mathrm{m}$ high, the optical sensor should be at least $3\,\mathrm{m}$ from the edge in any direction it can see.

  – Be sure to take all B readings at the same height and in same direction as the A reading.

**A2   Leaf chlorophyll concentration measurement protocol**

1. Select and tag 5 plants in each field for continuous measurements.

2. Tag the 5th and the 6th leaves of each of the 5 plants

3. On each leaf, take 3 measurements at 1/4, 1/2 and 3/4 of the distance from the leaf base to its tip.

4. Take the readings on both leaves (6 measurements per plant).

5. Press Average to generate the average SPAD reading for the plant

The SPAD readings were taken from plants within the four same sections where the LAI readings were taken, and from another section of the field (making 5 sections in all).

**Appendix B: Biomass and yield measurement protocol**


1. Measure quadrant size of $6\,\mathrm{m}$ x $6\,\mathrm{m}$ diagonally in 3 different location in the field.

2. Count the number of plants in each quadrant.

3. Count number of cobs harvested from each quadrant.

4. Determine the number of cobs per $m^2$.



5.  Remove the ears leaving the husk intact on plant.

6.  Weigh the total cobs harvested from each quadrant separately.

7.  Select 10 cobs, weigh and shell and weigh the grains.

8.  Take $500\,\mathrm{g}$ of the grain sub-samples from each quadrant for moisture content determination.

9.  Weigh all empty cobs and then $500\,\mathrm{g}$ taken from each sample for oven drying

10.  Label the samples from each quadrant clearly with the quadrant number. Put all the 3 samples from the farm into 1 polybag and label the bag with the name of the farmer, community, and district.

11.  Submit the grain sample for moisture content determination within 24 hours.

12.  With known moisture content, estimate the grain yield per $\mathrm{mm}$.

13.  Cut plants in harvested area just above ground.

14.  Select 10 representative plants (stover) into leaf blade, husk, leaf sheath and stem (including tassel).

15.  Weigh each component and log weights (undried). Chop components separately, take a sub-sample of $500\,\mathrm{g}$ of each of the undried component. Place in sampling bags and label.

16.  Oven dry each weighed undried sub-sample to a constant weight.

17.  Find a ratio of the dried to the undried sub-samples of each component and multiple by their respective total undried
weight to obtain dry weight of each component.

18.  Add all total dried weight of all components (leaf blade + leaf sheath + stem (including tassel and husk) + cob) to obtain total biomass.

*Author contributions.* Jose Luis Gomez Dans, Philip Lewis, Feng Yin, Dilys McCarthy and Kofi Asare designed the study and wrote the manuscript in close collaboration with the other authors. JH and XL contributed Section 2.7. *In situ* samples were collected by Martin Addi,
Caroline Edinam Doe, Stephen Aboagye-Ntow, Rahaman Alhassan and Kenneth Kobina Yedu Aidoo, Patrick Lamptey and Isaac Kankam Boadu supervised the collected samples. All co-authors contributed to data evaluation and interpretation and editing of the manuscript.

*Competing interests.* The authors declare no competing interests.



*Acknowledgements.* The authors would like to acknowledge support from the Newton Prize (Newton Prize 2019 Chair's Award, administered by BEIS), the United Kingdom's Natural Environment Research Council (NERC) National Centre for Earth Observation (NCEO) NC ODA
Full project (NE/R000115/1) and STFC under project AMAZING- Advancing MAiZe INformation for Ghana (ST/V001388/1). The 2020 campaign was funded by NERC NCEO under NC ODA Full project (NE/R000115/1). The Newton Prize funded the purchase of equipment for this work, as well as the personnel costs for the 2021 campaigns. We acknowledge the Planet Education and Research Program, for providing access to the Planet data used in this study.



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
