# Peer review of "Location, biophysical and agronomic parameters for croplands in Northern Ghana"

_Earth System Science Data, 2022_

## Author Comment (AC1)

**Response to the reviewers**

We thank the reviewer for taking the time to go over our work. In the following, we address their concerns point by point.

Where required, we also provide excerpts of the manuscript (MS) with removed text marked in red and crossed over, new text in blue and underlined.
* * *
**Reviewer 1**

**Reviewer Point P 1.1** — One major source of uncertainty is the selection of data products. For instance, the authors choose PlanetScope surface reflectance data to derive the vegetation indices, which is then used to estimate LAI. They mention that there is a lot of noise in this dataset, which requires a couple of rounds of outlier detection and removal and that the positional accuracy can be as large as 10 m. If so, why not used Sentinel surface reflectance to estimate NDVI? Sentinel has a regular acquisition time, lower positional errors, and more consistency of atmospheric corrections. The final dataset is at around 10 m resolution, so I am unsure about the added benefit of Planet imagery here.

**Reply**: The rationale for using PlanetScope is due to the insufficient sampling of Sentinel 2 due to persistent cloud. To make this clear, we have added a new Fig. 5 to show the observational opportunity. We have also checked that the NDVI trends are consistent with Sentinel 2, and added the Sentinel 2 field averaged NDVI observations for reference in (now) Fig. 6. We believe that these updates clarify the reasoning behind using Planet data instead of Sentinel 2. Section 2.5 now reads

> *Together with the ground data described above, we have also produced an analysis-ready dataset (ARD) a ready-processed dataset of contemporaneous satellite observations to facilitate training and experimentation for dissemination. We . We will use the ground data to develop an empirical estimation of LAI. We have used the Planet Surface Reflectance (SR) version Fig. shows time series of the field averaged NDVI over four fields derived from Sentinel 2 product (Planet, 2018) to give sufficient temporal sampling over the crop season, as cloud cover prevalent during the rainy/growth season limits the use of other optical data such as from Sentinel-2 or Landsat for much of that time. The goal here is to demonstrate the application of and Planet. Sentinel 2 observations are restricted by clouds, whereas Planet data are more frequent, particularly towards the end of the growing season. When both sensors collect data, the field datato calibrate a simple NDVI to LAI model to provide a spatial estimate of LAI for each sample field and extend the LAI dataset. We choose NDVI as the surrogate for mapping LAI NDVI value is comparable, although it is also clear that the Planet data show a larger instability in time, as well as the presence of outliers. It would have been preferable to use Sentinel 2 observations to produce estimates of LAI as the data have a richer spectral information content, but given the scarcity of match ups with the ground measurements, we decided to use the Planet data, and to develop a simple mapping using a vegetation index as a pragmatic trade off.*

> *We have used the Planet Surface Reflectance (SR) version 2 product (Planet, 2018) downloaded from Planet Explorer (`https://www.planet.com/explorer/`). We calculate NDVI as it is a commonly used vegetation index that is frequently used to describe crop condition and yield (Turner et al., 1999; Smith et al., 2002; le Maire et al., 2004; Ferwerda and Skidmore, 2007; Le Maire et al., 2008).*

> *Surface reflectance data were subset and downloaded from Planet Explorer (). This The Planet SR product is derived from the top of atmosphere (TOA) radiance images acquired by the PlanetScope constellation which collects data in the red, green, blue and near infrared bands with a nominal resolution of ∼3.7 m. The SR product has a ground sampling distance of ∼3 m and a positional accuracy better than 10 m (Planet, 2018). The data are atmospherically corrected and have an associated cloud, cloud shadow, etc. pixel mask (Planet, 2018). Even so, the*

> *The vast changes in acquisition geometry, sensor properties, failure of the cloud and cloud/shadow mask and inconsistencies in the atmospheric correction result in the measurements from Planet being very noisy and contaminated with outliers., as is clear from Fig. (see also Houborg and McCabe (2016)*

*). Outliers and gaps in the time series (particularly at the start of the measurements period) require treatment: we develop here a robust smoothing and interpolation approach that allows us to achieve the desired NDVI to LAI mapping, along with an estimate of LAI uncertainty.*

*We use an efficient and robust smoothing filter with a bi-square weighting to flag and remove gross outliers in the Planet NDVI time series (Heiberger and Becker, 1992; Garcia, 2010). An outlier is flagged if $\left|\frac{u_i}{4.685}\right| \geq 1$, where $u_i$ is the studentised residual for sample $i$ (Garcia, 2010). An example application of the smoother is shown in Fig. (a), where the Sentinel 2 field averaged NDVI is also shown for comparison.*

*[…]*

*We use the interpolated and smoothed NDVI data to develop the mapping to LAI. A potential The large uncertainty in the individual elemental sampling unit (ESU) LAI ground measurements suggests that the model is fitted at field level. A potential further issue with a mapping from NDVI to LAI are saturation effects with high LAI (Baret and Guyot, 1991). For maize in the study area, very high LAI is never achieved, and the field measurements never exceed an LAI of 3, so we might suppose that saturation of the signal should not be a problem here. The limited range of the field data LAI data also suggests that a linear model is an acceptable model choice. We estimate the value of NDVI on the day of the in situ observations from the smoothed/interpolated Planet data, and average both the EO estimated NDVI and the in situ LAI over the field. We randomly split the data set set into $70\%$ for training and $30\%$ validation. We fit the linear model $LAI = m \cdot NDVI + c$ to the training data and test its performance on the validation samples. We repeat this fitting procedure using twenty random splits to avoid biases in the estimates of $m$ and $c$ and to provide an initial uncertainty on these parameters.*

[Figure]

Figure 5 : Field averaged NDVI from Sentinel 2 (green dots) and Planet (purple squares) over four of the visited maize fields in 2021. Vertical purple lines indicate the extent of the *in situ* data gathering campaign. Error bars indicated 2-98% field NDVI percentiles.

**Reviewer Point P 1.2** — Another source of uncertainty related to data products is the landcover mask used based on the ESRI global 10 m land cover dataset. Why was this classification dataset chosen instead of other similar 10 m land cover datasets (Venter et al. 2022)? Was the same ESRI landcover classification used for both years or did the authors use the 2020 and 2021 land cover products (https://planetarycomputer.microsoft.com/dataset/io-lulc-9-class) separately? I am concerned how using these difference datasets would impact the final results and datasets produced.

[Figure]

Figure 6 : Planet VI time series processing steps example for field 7074ZIN. (a) Outlier filtering. (b) First pass single pixel double logistic fitting (unconstrained). (c) Second pass single pixel double logistic fitting (phenology parameters constrained by median field). The vertical lines show the extent of the ground campaign period.

**Reply**:  Thank you for your question and the suggested reference. We used the ESRI global 10m land cover data in 2020 in this study. We checked the data we used for crop mask and found these products are very similar within our study area. Although the ESRI map is the best in three products in a comparison study (Venter et al., 2022), all these products show similar performance in Northern Ghana. Meanwhile, the inter-annual land cover dynamics of the ESRI map show only minor variation from year to year over the region of interest. Therefore, we think the crop map we used in study is reasonable and would not considerably impact the derived results.

Additionally, the code for the maize mask (provided in the paper) can be modified by users to user other base maps. We have added some text to reflect that in Section 2.7:

> *After applying a cloud mask, and only processing pixels labelled as "Crops" in the ESRI Sentinel 2 landcover map (Karra et al., 2021)(although the code provided is flexible and users can modify the base map and its classes easily), temporal series of number of vegetation*

**Reviewer Point P 1.3**  —  The second source of uncertainty is regarding the methods used and how they are impacting biases in the final dataset. One concern is about the outlier detection. This is done in a more or less statistical manner. However, is there a way to check with the in situ observations whether the outliers are a real signals or noise. Here, it would also be good to see the vegetation index from Sentinel in Fig. 5. If the outlier is purely due to the uncertainties in the PlanetScope estimates, it might be better to use Sentinel for calculating the NDVI?

**Reply**:  This question deals with the impact of the outlier rejection described. This step is needed because of imperfect cloud and cloud shadow masking, which is an issue with any optical EO sensor. We have addressed the rationale for using Planet data over Sentinel 2 in point P 1.1 already.

Using the collected field data to flag outliers is challenging, as data were collected once per week, and they do not cover the entire growing period (see Fig. 11).

We have added some extra text qualifying the outlier filtering in Section 2.5 in addition to the changes highlighted in point P 1.1:

> *Although the outlier filtering method described above is based on smoothing and statistical tests, the spatially-aware field constraints and typical consistency in reconstructed VI trajectories (Fig. (c)) over a field suggest that the outlier filtering is appropriate, and does not introduce large biases. The processing described above results in more stable estimates of NDVI over time, as can be seen in Fig. (c), particularly tightening up the temporal trajectory towards the start of the time series.*

**Reviewer Point P 1.4** — The overall accuracy of the derived LAI and the NDVI to LAI are both quite low (Fig. 9) with a correlation coefficient of 0.49 (so r2 of around 0.25). Is this a reasonable accuracy for such a dataset and how would end users justify using this dataset if such a low proportion of the variance is being explained? Here, I am also surprised why the authors showed the r value in Fig. 9 and the r2 value in Fig. 14. Best to be consistent.

**Reply**: We have added some more comments and comparisons with the literature, and partly re-written Sections 3.3 and 4.2:

> *The approach described in Sect 2.5 results in a simple transformation between Planet NDVI and LAI. The calibration and validation of this approach are shown in Fig. . The*  *conversion equation is given by $LAI_{pred} = 3.95 \cdot NDVI - 1.21$, with the two coefficients have bootstrapped uncertainties of 0.16 and 0.09, respectively. In validation, the model shows a modest correlation ($R = 0.5$, $R^2 = 0.25$), but in absolute terms, the model performs in line with medium resolution products (Fang et al., 2019), with a validation root mean squared error (RMSE) around*  $0.43\,\mathrm{m}^2\mathrm{m}^{-2}$, *mean absolute error (MAE) was* $0.35\,\mathrm{m}^2\mathrm{m}^{-2}$, *and negligible bias (Fig. ). Fig. clearly shows an underestimation of the Planet NDVI signal for $LAI > 1.5$.*  *A comparison of the field LAI measurements and the Planet-derived LAI time series is presented in Fig. 11, where a correspondence between the model predicted LAI and the field measurements is shown.*

Also:

> *We presented a very simple approach to exploit the ground data set and calibrate a simple LAI relationship to satellite NDVI. The outlier filtering and logistic function fitting are fairly standard approaches, and are necessary pre-processing stages to establish a linkage between ground and satellite measurements. While the results of the mapping appear in line with similar approaches*  *(e.g. Fig. 4 in Fang et al. (2019) shows a median RMSE for LAI around 0.5, although larger correlations are typical), and better than universal relationships for maize (Kang et al. (2016) report an RMSE*  $\sim 1\,\mathrm{m}^2\mathrm{m}^{-2}$ *for maize, against our reported* $\sim 0.5\,\mathrm{m}^2\mathrm{m}^{-2}$), *there are issues with poor correlation and high LAI samples in Fig. .*  *The small dynamic range of the field data, coupled with the large uncertainties of the measurements are behind both of these effects. Ground uncertainties arise from measurements in a heterogeneous, sparse and discontinuous canopy, whereas the low variability of the ground data is caused by the period of data gathering not providing a full description of the entire vegetation growth dynamics (see Fig. 11). A further potential source of uncertainty is the contribution of the soil to the NDVI signal (Baret and Guyot, 1991; Carlson and Ripley, 1997).*
>
> *The evaluation metrics presented here and the suitability of this data and method have to be evaluated for particular applications. In some applications, the low bias estimate of LAI and acceptable RMSE performance of the model will make this data useful, whereas for others, just using the filtered and smoothed NDVI trajectory may be more appropriate.*

**Reviewer Point P 1.5** — Looking at Fig. 10, there are both systematic biases and differences in phenology between predicted and field LAI. Is this bias somehow incorporated in the final dataset? It would be helpful to include some indication of this bias so that end users know what the uncertainties are over a field before they use the results.

**Reply**: The bias and uncertainties have been evaluated and presented in an updated Fig. 10. See also the text changes in Section 3.3 amended in point P 1.4.

[Figure]

Figure 10 : NDVI to LAI calibration (grey lines show bootstrap uncertainty) (left) and validation (right)

**Reviewer Point P 1.6** — How do this dataset addresses the issue initially raised in the introduction. As an example, in the introduction, the authors talk about the limitations of remote sensing due to the presence of trees, inter-cropping practices, etc. But then they choose the fields with the least amount of tree cover and inter-cropping for the in situ crop measurements. I think the authors need to expand upon this discussion or modify the introduction.

**Reply**: We chose the "cleaner" fields because none of the usual EO-driven biophysical parameter retrieval schemes deal with mixed cropping or different canopy layers. We have added a clarification in the text (Section 2.3), and have also extended the introduction to suggest uses of the data

> *The crops in the fields surveyed in this data set are grown by smallholder farmers and represent a typical sample of the variability found in this region, and provide a strong foundation for developing and assessing land cover or crop type maps. The crop biophysical and agronomic parameters provide an important source of data to develop and adapt crop monitoring methods to typical West African conditions, validate satellite-derived estimates of important biophysical parameters, as well as a useful source of data to validate the performance of crop growth models parameterised for maize in the region using typical fields.*

> *As for the crop mapping, the selected fields show no intercropping, and the presence of trees is limited to the edges and have been masked out. These decisions limit the selection of fields, but provide a simpler setting to validate EO-derived products and to test the link between biophysical parameters and crop production. Heterogeneous fields required different measurement strategies to characterise the nature of the crop combination, and the presence of several canopy layers (e.g. crop-tree) is not considered in most EO LAI products, so tree detection and masking using VHR data would be needed to make any comparison fair.*

**Minor points**

**Reviewer Point P 1.7** — It is unclear how comprehensive this dataset is. What is the fraction of the total area of the smallholder maize fields in Ghana that this dataset pertains to?

**Reply**: The dataset size is mentioned in the abstract. The crop type location data set is quite large (1800 fields across three agroclimatic zones) to be useful for developing and testing landcover applications. The biophysical parameters have been collected repeatedly over nearly 50 typical maize fields, and would be representative of maize grown in the Guinea savanna region of Ghana. We have added the following text to Section 2.3, and have also extended the introduction to suggest uses of the data (see point P 1.6

> *These fields are representative of typical maize fields grown in the Guinea savanna region of Ghana.*

**Reviewer Point P 1.8** — Line 115: Here and elsewhere, probably best to be explicit that these are in Celsius.

**Reply**: Fixed. Thanks!

**Reviewer Point P 1.9** — Line 200: saturation effects with high what?

**Reply**: ... LAI. Fixed. Thanks!

**Reviewer Point P 1.10** — Line 230: How were the pixels split? Randomly? Some kind of stratification? Was there only one set of training/validation? Why not use multiple random splits to check for consistency of results?

**Reply**: Thanks for this suggestion. They were split randomly, and we now report results from running 20 random splits and showing bootstrap uncertainty in the different parameters. Added in Section 2.6 (and uncertainties also reported in Section 3.3 and Fig. 10).

> *We repeat this fitting procedure using twenty random splits to avoid biases in the estimates of $m$ and $c$ and to provide an initial uncertainty on these parameters.*

---

## Author Comment (AC2)

**Response to the reviewers**

We thank the reviewer for taking the time to go over our work. In the following, we address their concerns point by point.

Where required, we also provide excerpts of the manuscript (MS) with removed text marked in red and crossed over, new text in blue and underlined.
* * *
**Reviewer 2**

**Reviewer Point P 2.1** — * ln 20: the link to the data description paper is broken

**Reply**: We have checked this link, and found it working and pointing to the right repository.

**Reviewer Point P 2.2** — * ln 230: how were the splits performed? how were features selected for in the random forest classifier to avoid overfitting?

**Reply**: We have now reported the average overall accuracy resulting from a 5-fold cross-validation to test the robustness of the reported statistics. We also now include a figure showing the feature importance for the classifier, and a brief discussion in Section 4:

(Section 3.5)

5-fold crossvalidation was also used to evaluate the robustness of the results shown in Table 2, which resulted in an overall accuracy of 0.72 (standard deviation 0.12). The importance of each considered features is shown in the Gini index plot shown in Fig. 15, which ranks different features by classifier importance.

Section 4.3

The maize mask that was developed in this paper demonstrates that the data can be used as an input to a classifier. However, the limited number of samples for 2021 (where the main aim of the field campaign was biophysical parameter collection) result in a crop mask that is probably only reliable around the collected data points. Also, since the surveyed fields were selected as late sown, this may also bias the field selection. Fig. 15 indicates that the classifier is mostly being driven by observations around the first half of June (DoYs 150-165), suggesting that early crop development may be more informative for crop discrimination than late crop development.

**Reviewer Point P 2.3** — *ln 445: the link to the code for classification seems broken or unavailable

**Reply**: Fixed, thanks!

**Reviewer Point P 2.4** — * ln 455: In situ biophysical parameter time series csv file does not seem available at the zenodo page provided.

**Reply**: The biophysical parameters are stored in file `Ghana_ground_data_v5.csv`. This file has been part of the data set, but as a way of clarifying where data are, we have added the filenames to Section 7.

**Reviewer Point P 2.5** — The dataset lacks a metadata which could be useful to decribe the data for users who may want to explore the data further.

**Reply**: We hope that this publication acts as a thorough description of the data set and a list of possible uses, and the dataset will be submitted to MLHub (`https://mlhub.earth/`), to give it wider visibility.